# Gut Bacterial Dysbiosis in Children with Intractable Epilepsy

**DOI:** 10.3390/jcm10010005

**Published:** 2020-12-22

**Authors:** Kihyun Lee, Namil Kim, Jung Ok Shim, Gun-Ha Kim

**Affiliations:** 1Chunlab, Inc., ChunLab Tower, 34 Samseong-ro 85-gil, Seoul 06194, Korea; kihyun.lee@chunlab.com (K.L.); namil.kim@chunlab.com (N.K.); 2Division of Pediatric Gastroenterology, Hepatology, and Nutrition, Department of Pediatrics, Korea University College of Medicine, Seoul 08308, Korea; shimjo@korea.ac.kr; 3Division of Pediatric Neurology, Department of Pediatrics, Korea Cancer Center Hospital, Korea Institute of Radiological & Medical Sciences, Seoul 01812, Korea

**Keywords:** microbiome, microbiota, intestinal flora, stool, epilepsy, ketogenic diet

## Abstract

A few published clinical studies have evaluated the association between gut microbiota in intractable epilepsy, but with inconsistent results. We hypothesized that the factors associated with the gut bacterial composition, such as age and geography, contributed to the discrepancies. Therefore, we used a cohort that was designed to minimize the effects of possible confounding factors and compared the gut microbiota between children with intractable epilepsy and healthy controls. Eight children with intractable epilepsy aged 1 to 7 years and 32 age-matched healthy participants were included. We collected stool samples and questionnaires on their diet and bowel habits at two time points and analyzed the gut microbiota compositions. In the epilepsy group, the amount of Bacteroidetes was lower (Mann–Whitney test, false discovery rate (FDR) < 0.01) and the amount of Actinobacteria was higher (FDR < 0.01) than in the healthy group. The epilepsy subjects were 1.6- to 1.7-fold lower in microbiota richness indices (FDR < 0.01) and harbored a distinct species composition (*p* < 0.01) compared to the healthy controls. Species biomarkers for intractable epilepsy included the *Enterococcus faecium* group, *Bifidobacterium longum* group, and *Eggerthella lenta*, while the strongest functional biomarker was the ATP-binding cassette (ABC) transporter. Our study identified gut bacterial dysbiosis associated with intractable epilepsy within the cohort that was controlled for the factors that could affect the gut microbiota.

## 1. Introduction

The microbiota–gut–brain axis has received immense attention from the research community over the past few years. This axis signifies that the gut microbiota and the brain can communicate in both directions through the nervous system and endocrine, immune, and metabolic pathways. Recent studies have reported that changes in the gut microbiota are associated with diseases of the central nervous system, including autism spectrum disorder [1,2], multiple sclerosis [3], Parkinson’s disease [4,5], and Alzheimer’s disease [6].

Epilepsy is a non-communicable neurological disease that affects all ages, where the recent global burden was estimated to be 45.9 million patients and 126,055 epilepsy-related deaths per annum [7]. Heritable and acquirable risk factors linked to the genetic and epigenetic mechanisms of epilepsy have been actively identified for decades [8]. The variants that were discovered to be associated with epilepsy were frequently linked to cardiac ion channels [9], which led to the concept of a genetically determined cardiocerebral channelopathy, with an implication of the link between the genetic basis of epilepsy and cardiovascular diseases [10].

Apart from the advancements made in the understanding of the genetic mechanisms underlying epilepsy, a line of evidence has emerged in recent years that suggests a link between the gut microbiota and seizure control [11].

The association between antibiotics and seizures has been postulated in several case reports [12], suggesting the role of the gut microbiota in seizure control, while other studies have reported the role of antibiotics in inducing significant alterations in gut bacterial communities [13]. One case report described a patient who received fecal microbial transplantation for the treatment of inflammatory bowel disease, which unexpectedly resulted in the remission of the concomitant epilepsy [14]. 

Notably, animal experiments have demonstrated that gut microbes are needed to mediate the effects of the ketogenic diet, which is a remarkable treatment strategy for intractable epilepsy. For example, one study found that the anticonvulsant effect of the ketogenic diet disappeared when the gut microbiota of mice was depleted by antibiotics or aseptic breeding and that the anticonvulsant effect of the ketogenic diet could be “transmitted” to the control diet mice via fecal microbial transplantation [15].

In recent years, several clinical studies have reported the differences between the microbial distribution in patients with intractable epilepsy and healthy controls, as well as microbial changes after the ketogenic diet. However, the results of these studies were inconsistent (see Table 1 for comparison). We assumed that the failure to consider factors that could affect the distribution of gut bacteria, such as age, body mass index, modes of delivery, type of diet, and recent antibiotic exposure, in the study design was among the reasons responsible for this discrepancy. Therefore, we sought to minimize the effects of these factors while comparing the gut microbiota of children with intractable epilepsy and healthy controls in this study.

## 2. Experimental Section

### 2.1. Study Design

This prospective study included patients with intractable epilepsy and healthy participants who were recruited from Korea University Guro Hospital in Seoul, Korea, between March 2018 and September 2018. The inclusion criteria for the patient group were as follows: patients who (1) were aged 1 to 7 years; (2) had intractable epilepsy with uncontrolled seizures, despite the administration of two or more antiepileptic drugs; (3) had no history of chronic gastrointestinal disorders; (4) had not used antibiotics within the last 4 weeks. Age-matched healthy volunteers who did not have any symptoms of a chronic gastrointestinal disorder or a history of chronic illness and had not taken antibiotics within the last 4 weeks were recruited as controls. We collected clinical data about the age, height, weight, modes of delivery, type of diet, and recent antibiotic exposure from both groups of participants.

We obtained written consent from all participants and their parents prior to enrollment in the study and sample collection. Each patient or healthy volunteer received a stool sampling kit (SB-01; ChunLab Inc, Seoul, Korea) including instructions, a sterile collection tube with a preservation buffer, an ice pack, and insulated thermal mailers. The participants collected their stool samples at home following the instructions. Once the fecal samples had arrived in the lab, they were frozen at −80 °C until DNA extraction for 16S RNA gene sequencing. Their parents completed a questionnaire for participants’ food intake and bowel movements. The institutional review board of Korea University Guro Hospital (IRB No.: 2018GR0120) approved this study.

### 2.2. 16S rRNA Gene Sequencing

The fecal samples were thawed, homogenized in suspension, bead-beaten, and centrifuged at 14,000× *g* for 10 min. The supernatants were diluted with nuclease-free water and used as templates for polymerase chain reaction amplification. The hyper-variable region (V3–V4) of the bacterial 16S rRNA was amplified using 341F and 805R primers [20]. Amplicons were pooled in equal proportions and purified using an AMPure XP purification kit (Beckman Coulter, Indianapolis, IN, USA) according to the manufacturer’s instructions. Purified amplicon libraries were sequenced at ChunLab Inc. (Seoul, Korea) using the Illumina MiSeq platform with MiSeq Reagent Kit v3 (Illumina, San Diego, CA, USA) with 2 × 250 bp paired-end cycles.

### 2.3. Taxonomic Comparison and Diversity Indices

The taxonomic profile of the bacterial community was analyzed using EzBioCloud’s Microbiome Taxonomic Profiling cloud using the database version PKSSU4.0, as described in a previous study [16]. The PKSSU4.0 database classifies each group of species sharing an average nucleotide identity of ≥95% in the “species group,” as their members are not distinguishable from each other under the sequence identity thresholds used in 16S rRNA gene-based taxonomic profiling surveys. The biomarkers discovered in this study included some of these species groups (e.g., the *Enterococcus faecium* group). The full list of conventional species names of each species group can be found on the EzBioCloud taxonomy webpage (https://www.ezbiocloud.net/mtp/taxonomy).

We normalized the read counts to 22,463, which was the smallest read count achieved among all the samples, to compare the α-diversity metrics of the samples. We applied normalization based on the 16S rRNA gene copy number variation for compositional variation (i.e., β-diversity metrics) and biomarker discovery analyses [21,22]. 

The species richness was assessed using the Chao1, abundance-based coverage estimator (ACE), and Jackknife estimators, and using the numbers of observed operational taxonomic units (OTUs). Diversity indices were calculated using the NPShannon, Shannon, and InvSimpson estimators, and were based on the phylogenetic diversity using the OTU occurrence matrix. Variations in the OTU profiles in the samples were quantified using the generalized UniFrac metric and visualized with hierarchical cluster trees using the unweighted pair group method with the arithmetic mean (UPGMA) and principal coordinate analysis (PCoA). The phylogenetic investigation of communities via the reconstruction of unobserved states (PICRUSt) algorithm was used to determine the functional potentials of the bacterial communities at the Kyoto Encyclopedia of Genes and Genomes ortholog, pathway module, and pathway levels.

The associations between the variation in taxonomic profiles and diversity indices and the sample category, i.e., the epileptic or healthy state, were analyzed using the Mann–Whitney *U* test and permutational multivariate analysis of variance (PERMANOVA). We applied the Benjamini–Hochberg adjustment to calculate the false discovery rates (FDRs) while testing multiple variables (e.g., taxa) against the same predictor variable (e.g., the sample group). The linear discriminant analysis effect size (LEfSe) was employed to determine the taxa and inferred pathways that were strongly associated with one of the two sample groups in order to identify the potential biomarkers. We used a stringent logarithmic linear discriminant analysis (LDA) score threshold of 4.0 for taxonomic markers and 3.0 for functional markers for the LEfse outputs.

## 3. Results

### 3.1. Study Population

This study included 8 children with intractable epilepsy and 32 healthy controls. The baseline characteristics of the patients with intractable epilepsy are shown in Table 2. The patient’s age was 38.0 (25.3; 56.0) months (median (interquartile)) and the female-to-male ratio was 5:3. Two patients were on a liquid diet: one patient was fed an artificial feeding formula, and the other was provided enteral nutrition via a gastrostomy tube. 

The control group included 16 stool samples from healthy volunteers and 16 sample data procured from ChunLab. We compared clinical data that could possibly act as confounding factors between patients with intractable epilepsy and healthy controls, where the results are summarized in Table 3. We found no differences in body mass index, modes of delivery, the number of children on a special diet, or the presence of constipation between the two groups.

### 3.2. Taxonomic Compositions

We generated 53,090 ± 31,692 reads of bacterial 16S rRNA genes per sample and estimated the taxonomic composition and diversity in the stool of children with epilepsy and the normal controls. Firmicutes were the most common phylum (i.e., taxon) in both groups (Figure 1a). The proportion of Bacteroidetes was lower (Mann–Whitney *U* test, FDR < 0.01) and that of Actinobacteria spp. was higher (FDR < 0.01) in the epilepsy group than in the healthy group. The proportion of Verrucomicrobia was higher in the epilepsy group, while that of Proteobacteria was higher in the healthy group; however, these respective differences lacked statistical significance (FDR > 0.1). At the family level (Figure 1b), the proportions of *Actinomycetaceae*, *Bifidobacteriaceae*, *Coriobacteriaceae*, *Corynebacteriaceae*, and *Enterococcaceae* were higher in the epilepsy group than in the control group (FDR < 0.05).

### 3.3. Alpha Diversity

We also analyzed the alpha diversity (species diversity within a single sample). The microbial richness was assessed using the number of OTUs and ACE, Chao1, and Jackknife indices after normalizing the read counts across samples (Figure 1c). The microbial richness was 1.6- to 1.7-times higher in the healthy control group than in the epilepsy group (FDR < 0.01), irrespective of the index used. The biodiversity index was assessed using the NPShannon, Shannon, and Simpson indices, as well as phylogenetic diversity, as shown in Figure 1c. Each index suggested that the level of community diversity was higher in the healthy group than in the epilepsy group (FDR < 0.01).

### 3.4. Beta Diversity

The variation in the species-level composition of the stool samples of healthy participants and those with epilepsy was assessed based on UniFrac distances. Principal coordinates (Figure 1d) and the results of UPGMA clustering (Figure 1e) indicated that patients with epilepsy harbored distinct species compositions compared to their healthy counterparts. The PERMANOVA supported the finding that the community structure in the epilepsy patients showed a significant association with the variation in species composition (*p* = 0.001).

### 3.5. Effect of Constipation and a Liquid- or Formula-Based Diet

We measured the effect of possible confounding factors, chronic constipation, and feeding in liquid form in our comparison between the epilepsy and control groups. Based on questionnaires, the number of children with constipation was not different between the two groups (Table 3). Species diversity estimates were not correlated with the presence of constipation within the epilepsy group (Appendix A), and the taxonomic composition was also independent of the presence of constipation (Appendix A). Two subjects in the epilepsy group were on a liquid-form diet, where one child was fed through a gastrostomy tube and the other was taking an artificial formula. We repeated the taxonomic composition, species alpha- and beta-diversity analyses excluding these two subjects, and the results were mostly identical to the original results (Appendix A).

### 3.6. Taxonomic and Functional Biomarker Discovery

We identified 17 and 18 bacterial taxa that were strongly associated with the epilepsy group and healthy control group, respectively (Table 4). At the species level, the *Enterococcus faecium* group, *Bifidobacterium longum* group, and *Eggerthella lenta* were putative biomarkers of the epilepsy group, whereas the *Faecalibacterium prausnitzii* group and *Bacteroides vulgatus* were biomarkers of the healthy group (LEfSe; LDA effect size > 4.0, FDR < 0.05). 

We identified eight modules and three pathways that were strongly associated with the epilepsy group and a single module that was associated with the healthy group (LEfSe; LDA effect size > 3.0, FDR < 0.05) from among the PICRUSt-inferred pathways and modules.

## 4. Discussion

Several clinical studies on gut microbiota in intractable epilepsy reported inconsistent results, as summarized in Table 1. We presumed that the lack of consensus could be the result of inconsistency in their control over the confounding factors, such as age [23,24], body mass index [25], type of diet [26,27,28,29,30], modes of delivery [31,32], or recent antibiotic exposure [33], which are known to have an influence over the gut microbiota. Antibiotic exposure was the only variable that was consistently controlled throughout most of the previous studies. The effect of age was not controlled for in the previous studies, as they either did not match the age between the epilepsy and control groups (i.e., the parents of the patients were used as the control group) [16], included participants with too diverse of an age span (i.e., from infants to adults) [18,19], or did not include a control group [17]. The first year of life is known as a period of substantial fluctuation and maturation in the intestinal microbiome [24], and our unpublished preliminary study of healthy children showed that the species compositions before the age of 1 year were distinct from those after this age (Appendix A). In this study, we narrowed the age spectrum down to preschoolers aged 1 to 7 years in the patient and control groups and confirmed the lack of any statistically significant difference in the ages of both groups (Table 3). Moreover, we excluded participants who had recently been exposed to antibiotics and accounted for differences in body mass index, modes of delivery, and diet. 

We found a reduced Bacteroidetes abundance, a reduced alpha diversity, and a noticeably shifted community structure in the gut microbiome of children with intractable epilepsy compared to their healthy counterparts. The bacterial community properties that we found to be associated with epilepsy overlapped with the typical attributes in an immature gut microbiome: lower diversity and lower Bacteroidetes abundance have together been described as characteristics of a relatively immature gut microbiome within the context of age (associated with younger patients) [34], birth mode (associated with cesarean sections) [35], or antibiotic exposure (associated with the exposed) [36]. A lower prevalence of Bacteroidetes in epilepsy patients was previously pointed out in the results of Peng et al.’s [18] and Xie et al.’s [19] research. Reduced gut bacterial alpha diversities were pointed out in Lindefeldt et al.’s and Xie et al.’s reports [16,19]. Taken together, the features of gut microbiota structures in children with epilepsy are highly likely to represent a dysbiotic state. Low diversity dysbiosis is thought to be linked to the altered profiles of bacterial metabolites and the interaction with the host immune system. As an example, short-chain fatty acids, including butyrate produced by intestinal bacteria, have both local and systemic anti-inflammatory effects, and the reduction of microbially produced short-chain fatty acids due to dysbiosis contributes to various pathological conditions [37].

We discovered several taxa as potential biomarkers for an epilepsy-associated gut microbiome, among which, the *Enterococcus faecium* group, *Bifidobacterium longum* group, and *Eggerthella lenta* were included as species-level biomarkers. Among them, *Eggerthella lenta* has been reported to inhibit digoxin, a cardiac glycoside, through the cardiac glycoside reductase operon’s products, and the quantification of this operon in the gut microbiome could predict inter-individual variation in the pharmacokinetics of digoxin [38]. These results suggest that it is necessary to look at drug resistance from the perspective of both the human and microbial genomes. Interestingly, in our anecdotal follow-up analysis of one of our patients’ gut microbiota after implementing the ketogenic diet for one month, his seizures were reduced by more than 50% and the three biomarker species decreased dramatically (*B. longum* group, 3.3-fold; *E. lenta*, 6.0-fold; *E. faecium* group, 440-fold), and the alpha diversity increased (Appendix A). Overall, 12 out of 17 epilepsy biomarker taxa decreased and 17 out of 18 healthy biomarker taxa increased after starting the ketogenic diet, suggesting that the gut microbiota alteration might be mediating the positive effect of the ketogenic diet. Alterations of gut microbiota during a ketogenic diet treatment of epilepsy has been characterized in previous studies (Table 1). The high fat content in the ketogenic diet could provide favorable conditions for *Bacteroides* that metabolize bile salts [39], thus explaining the increased *Bacteroides* or phylum Bacteroidetes seen in Zhang et al. [17] and Xie et al. [19]. 

We also found that the ABC transporters were significantly active in patients with drug-resistant epilepsy, similar to the results of Peng et al. [18]. ABC transporters (e.g., P-glycoprotein and multidrug-resistance-associated proteins) use adenosine triphosphate to continuously pump the drug out of the cell, thereby resisting the concentration gradient of the drug [40]. Thus, increased intestinal expression of ABC transporters may reduce the absorption of antiepileptic drugs. Several bacterial ABC transporters have been implicated in multidrug efflux [41].

It should be emphasized that our study only elucidated the associations rather than the underlying mechanisms toward the role of gut bacterial communities in the pathology of intractable epilepsy. Outside our study, however, there have been several advancements in the field in recent years supporting the causal relationship in the gut microbiota–epilepsy association. The intestinal microflora affects various bodily functions, such as gene expression, neurotransmitter metabolism, and numerous other neurophysiological processes [42]. Studies have found that various neurochemicals acting on the human body are secreted by gut bacteria [43]. The link between the gut and the brain via the vagus nerve, which is also known as the gut–brain axis, renders gut microbes particularly important in the context of epilepsy [11,42]. Recent evidence has shown that gut microbes mediate the effects of the ketogenic diet in mice, which is an effective treatment for intractable epilepsy [15]. The changes observed in the microbial community are associated with an elevation in the concentrations of the inhibitory neurotransmitter gamma-aminobutyric acid and decreased excitatory glutamate in the hippocampus [44], both of which are associated with seizure suppression.

Of course, it may be speculated that anticonvulsants may have caused dysbiosis by altering the gut microbiota in patients with epilepsy. Lamotrigine has been shown to inhibit ribosome production in *Escherichia coli* in an in vitro study [45] and it has been demonstrated that clonazapem can be metabolized by intestinal microflora in an animal study [46]. However, none of the 16 drugs in the N03A (Anatomical Therapeutic Chemical classification) subgroup containing antiepileptic drugs have proven antibacterial effects in human studies [47]. Given that evidence for a direct interaction between antiepileptic drugs and the gut microbiota is still lacking, more specific studies on the implications of these interactions are needed in the future.

The first limitation of this study was that the number of patients included was relatively small. To minimize the effects of the long-term use of antiepileptic drugs and long-lasting uncontrolled seizures, and to limit the diversity of dietary changes, the patient group was narrowed down to preschoolers aged 1 to 7 years. It would be intriguing to see future studies using a larger sample size or variable age and geographic settings to know how conserved the microbial attributes of intractable epilepsy are across ages and populations. Second, as in previous studies, detailed information about food intake was not available. Instead, the type of food (liquid or solid) and the route of the diet (oral or tube) were identified, as people with refractory epilepsy often fail to eat solids. Nonetheless, keeping a food diary is the ideal way to check each patient’s food composition and quantity, and should be considered in future studies.

## 5. Conclusions

We characterized the features of gut microbiota that are associated with epilepsy in children while considering the factors that may cause spurious positive or negative associations, and we have confirmed that patients with intractable epilepsy have gut bacterial dysbiosis. Further research is needed to elucidate the causal relationship between these changes in gut microbiota and drug resistance.

## Figures and Tables

**Figure 1 jcm-10-00005-f001:**
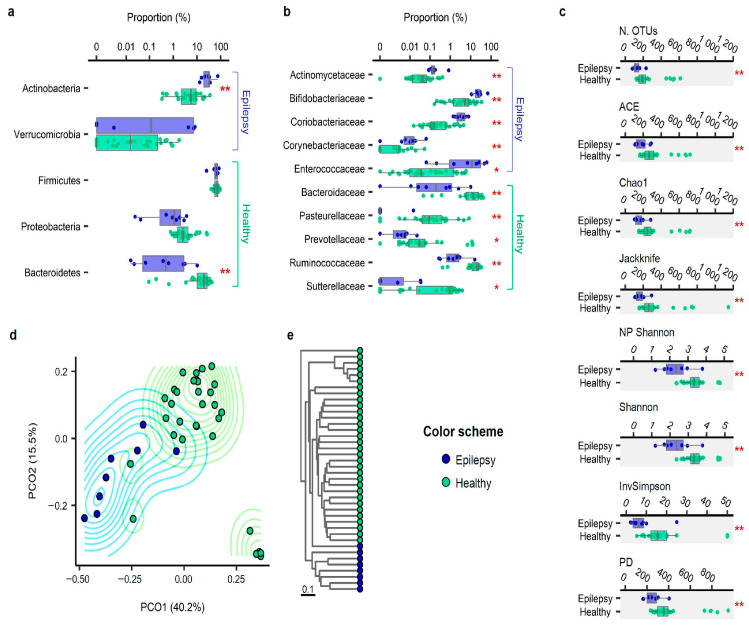
Comparison between the patients with intractable epilepsy and the healthy controls: (**a**) phylum level compositions of gut microbiota compared between the patients with intractable epilepsy and the healthy controls, where the taxa displaying significant differences are highlighted; (**b**) comparison at the family level; (**c**) species richness and diversity index compared between the patients with intractable epilepsy and the healthy controls; (**d**) principal coordinate analysis plot of the intersubject variation of bacterial-species-level operational taxonomic unit (OTU) compositions based on the UniFrac distance metric; (**e**) unweighted pair group method with the arithmetic mean (UPGMA) dendrogram representing the species-level compositional distances among the subjects. * labeled to the comparisons with a false discovery rate (FDR) < 0.05, ** *p* < 0.01, based on the Mann–Whitney test and Benjamini–Hochberg adjustment for multiple comparisons.

**Table 1 jcm-10-00005-t001:** Summary of previous studies on the intestinal microbiota in patients with intractable epilepsy.

Study	Population ^1^	Patients’ Age (Years) ^2^	Excluded or Matched Variables	Profile of Intractable Epilepsy Compared to the Controls ^1^				Change Before and After the Ketogenic Diet ^1^
				Taxon	α- Diversity	β- Diversity	Biomarker	
Our study	IE (*n* = 8), HC (*n* = 32)	3.17 (1.16–6.92)	Age, obesity, delivery methods, diet, antibiotic exposure	↓ Bacteroidetes, Proteobacteria ↑ Actinobacteria	Lower	SP	↑ *E.faecium*, *B.longum*, *E.lenta* ↑ ABCT	NA
Lindefeldt et al. [16]	IE (*n* = 12), Healthy parents (*n* = 11)	7.7 (2.2–15.3)	NA	NA	Lower	NA	↑ Hemin Transport system ↓ Carbohydrate metabolism	↑ *E. coli* ↓ Bifidobacteria, *E. rectale*, *Dialister* α-diversity: NS
Zhang et al. [17]	IE (*n* = 20)	4.2 (1.2–10.3)	Obesity, antibiotic exposure	NA	NA	NA	NA	↑ Bacteroides ↓ Firmicutes, Actinobacteria α-diversity: NS, β-diversity: SP
Peng et al. [18]	IE (*n* = 42), HC (*n* = 65), drug responsive epilepsy (*n* = 49)	28.4 (5–50)	Antibiotic exposure	↓ Bacteroidetes ↑ Firmicutes, rare species	Higher	SP	↑ ABCT ↓ Glucose and lipid metabolism	NA
Xie et al. [19]	IE (*n* = 14), HC (*n* = 30)	1.95 (0.8–3.3)	Antibiotic exposure	↓ Bacteroidetes ↑ Firmicutes	Lower	SP	NA	↑ Bacteroidetes ↓ Proteobacteria

^1^ Abbreviations: *E.faecium*, *Enterococcus faecium* group; *B.longum*, *Bifidobacterium longum* group; *E.lenta, Eggerthella lenta*; ABCT, ATP-binding cassette transporters; NA, not applicable, IE: intractable epilepsy, HC: healthy controls, NS, no significant change; SP, significantly separated. ^2^ Described as the mean or median mean (range). In the taxon and biomarker columns, the names of taxa and functional pathways were prefixed with upwards arrows (↑) when they are enriched in the epilepsy subjects, with downwards arrows (↓) when depleted in the epilepsy subjects.

**Table 2 jcm-10-00005-t002:** Baseline characteristics of the patients with intractable epilepsy.

No.	Age (Years)	Sex	Epilepsy Duration (Years)	Epilepsy Type	Etiology	Seizure Frequency	Past AEDs	Current AEDs	Type of Food
1	1.15	M	0.65	Infantile spasms	Unknown	>10 clusters/d	-	VGB, CLB, LVT, TPM	Artificial formula (oral)
2	4.81	F	4.31	Generalized	Unknown	3/d	-	VPA, LVT	Solid
3	2.63	F	2.05	Focal and generalized	Genetic	1/mo	LVT	VPA, CLB	Solid
4	4.52	F	3.44	Focal	Unknown	15/d	-	OXC, LVT	Solid
5	6.92	F	6.26	Focal and generalized	Unknown	3/d	VPA	VPA, PB, CLB, LVT, TPM, LTG	Enteral formula (tube feeding)
6	3.52	M	2.93	Generalized	Genetic	25/d	-	VPA, OXC, LTG, TPM	Solid
7	2.82	F	2.40	Infantile spasms	Structural	3–4 clusters/d	VGB, TPM	VPA, OXC	Solid
8	1.58	M	1.16	Infantile spasms	Structural	5–6 clusters/d	TPM	VGB, OXC	Solid

Abbreviations: M, male; F, female; AED, antiepileptic drug; VGB, vigabatrin; CLB, clobazam; LVT, levetiracetam; TPM, topiramate; VPA, valproic acid; OXC, oxcarbazepine; PB, phenobarbital; LTG, lamotrigine; d, day(s); mo, month(s); -, none.

**Table 3 jcm-10-00005-t003:** Comparison between the clinical data of the epilepsy and control groups.

Group	Epilepsy (*n* = 8)	Control (*n* = 32)	*p*
Sex ^1^			0.812
Female	5 (62.5)	16 (50.0)	
Male	3 (37.5)	16 (50.0)	
Age (years) ^2^	3.17 (2.11; 4.67)	3.67 (1.33; 5.27)	0.946
Body mass index ^3^	16.6 ± 2.3	16.4 ± 1.8	0.799
Mode of delivery ^1^			0.804
Vaginal	5 (62.5)	13 (76.5)	
Cesarean	3 (37.5)	4 (23.5)	
Liquid diet ^1^	2 (25.0), on a liquid diet ^4^	0	0.174
Constipation ^1^	3 (37.5)	5 (31.3)	1.000

^1^ Number (%), ^2^ median (interquartile range), ^3^ mean ± standard deviation. ^4^ One patient was fed an artificial formula and the other patient was fed an enteral formula via a gastrostomy tube.

**Table 4 jcm-10-00005-t004:** Comparison between the clinical data of the epilepsy and control groups.

Marker Type	Epilepsy Biomarkers		Healthy Biomarkers	
	Names	*N*	Names	*N*
Taxonomic ^1^		17		18
Species	*B. longum* group, *E. lenta*, *E. faecium* group	3	*Bacteroides vulgatus*, *Faecalibacterium prausnitzii* group	2
Genus	*Bifidobacterium*, *Eggerthella*, *Enterococcus*	3	*Bacteroides*, *Faecalibacterium*, *Lachnospira*, *Roseburia*, *Veillonella*	5
Family	*Bifidobacteriaceae*, *Coriobacteriaceae*, *Enterococcaceae*, *Streptococcaceae*	4	*Bacteroidaceae*, *Ruminococcaceae*, *Veillonellaceae*	3
Order	Bifidobacteriales, Coriobacteriales, Lactobacillales	3	Bacteroidales, Clostridiales, Veillonellales	3
Class	Actinobacteria_c, Bacilli, Coriobacteriia	3	Bacteroidia, Clostridia, Negativicutes	3
Phylum	Actinobacteria	1	Bacteroidetes, Proteobacteria	2
Functional ^2^		11		1
Module	Putative multiple sugar transport system (M00207), peptides/nickel transport system (M00239), energy-coupling factor transport system (M00582), putative ABC transport system (M00258), ABC-2 type transport system (M00254), PTS system, cellobiose-specific II component (M00275), PTS system, beta-glucoside-specific II component (M00271), putative aldouronate transport system (M00603)	8	Cobalamin biosynthesis, cobinamide to cobalamin (M00122)	1
Pathway	ABC transporters (ko02010), quorum sensing (ko02024), starch and sucrose metabolism (ko00500)	3		

^1^ Taxonomic biomarker selection criteria, linear discriminant analysis (LDA) effect size ≥ 4, FDR < 0.05; ^2^ Functional biomarker selection criteria, LDA effect size ≥ 3, FDR < 0.05. Accession numbers given to the functional modules and pathways are based on the Kyoto Encyclopedia of Genes and Genomes (KEGG) classification. Abbreviations: PTS, phosphotransferase system.

## Data Availability

The datasets generated during and/or analyzed during the current study are available from the corresponding author on reasonable request.

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
