# Peer review of "Gut Bacterial Dysbiosis in Children with Intractable Epilepsy"

_jcm, 2020, doi:10.3390/jcm10010005_

Round 1

Reviewer 1 Report

Authors describe that there are significant differences in gut bacterial compositions between pediatric patients with intractable epilepsy and age-matched healthy controls. In this manuscript, it is demonstrated that gut bacterial dysbiosis is associated with intractable epilepsy, and species biomarkers for intractable epilepsy are suggested. Gut bacteria has been considered to be related to intractable epilepsy in previous manuscripts. This manuscript may also provide some useful insight to understand pathophysiology of pediatric intractable epilepsy. However, there are some points to be ameliorated or clarified.

  • This study was conducted with quite a small number of patients. Authors should refer to limitation of this study from based on that point of view.
  • Contents of meals may paly crucial roles to compose gut bacteria plexus. Information of type and amount of nutrition in meals which participants had is needed to discuss intestinal microbiota.
  • Patients information is insufficient. At least authors should information of history of treatment and length of illness.
  • How do authors think about effect of AEDs on gut microbiota? Please describe that.
  • Contents of Acknowledgements section should be ameliorated.
  • In Table 2, Epileptic spasms is not appropriate for epilepsy type. It is one of seizure types.
  • In Table 2, “Concomitant AED” is not adequate. It should be changed to “Treatment (AED)” etc.

Reviewer 2 Report

The authors tried to clarify the association between gut microbiota in intractable epilepsy.

The topic is quite interesting but there are several issues to be defined:

  1. Introduction section should be expanded since the causal correlation between epilepsy and gut microbiota is something new for a reader. Moreover a very well isight into the literature should be given. I recommend these  references that cover the connection between the gut microbiome and epilepsy and the link between genetic basis of epilepsy and other diseases (such as cardiovascular implications):-Dahlin M, Prast-Nielsen S. The gut microbiome and epilepsy. EBioMedicine. 2019 Jun;44:741-746. doi: 10.1016/j.ebiom.2019.05.024. Epub 2019 May 31. PMID:31160269; PMCID: PMC6604367. -

     Coll M, et al. Genetic investigation of sudden unexpected death in epilepsy cohort by panel target resequencing. Int J Legal
    Med. 2016 Mar;130(2):331-9. doi: 10.1007/s00414-015-1269-0. Epub 2015 Sep 30.
    PMID: 26423924.

    Partemi S, et al  Genetic and forensic implications in epilepsy and cardiac arrhythmias: a case
    series. Int J Legal Med. 2015 May;129(3):495-504. doi:
    10.1007/s00414-014-1063-4. Epub 2014 Aug 15. PMID: 25119684.

  2. discussion and results sections: the discussion is poor and is not very well debated. Number of patients is very small, statistical not significant. These critical issues should be stressed out in the discussion. Moreover a limitation of the study should be acknowledged  at the end of the paper. Moreover the pathophysiology of the mechanism should be better defined.  

Round 2

Reviewer 1 Report

Authors addressed the comments raised. This manuscript was ameliorated.

Reviewer 2 Report

The authors did not included the references indicated in the previous review: these references are the proof of concept of the link between genetic basis of epilepsy and other diseases (such as cardiovascular implications) and should be included in the list of references: 

 Coll M, et al. Genetic investigation of sudden unexpected death in epilepsy cohort by panel target resequencing. Int J Legal Med. 2016 Mar;130(2):331-9. doi: 10.1007/s00414-015-1269-0. Epub 2015 Sep 30.PMID: 26423924. 

Partemi S, et al. Genetic and forensic implications in epilepsy and cardiac arrhythmias: a case series. Int J Legal Med. 2015 May;129(3):495-504. doi:10.1007/s00414-014-1063-4. Epub 2014 Aug 15. PMID: 25119684. 

Regarding the other issues raised in the previous review the paper is moderately improved.  
